## SCIENCE FORUM

# Adding a One Health approach to a research framework for minority health and health disparities

**Abstract:** The National Institute on Minority Health and Health Disparities (NIMHD) has developed a framework to guide and orient research into health disparities and minority health. The framework depicts different domains of influence (such as biological and behavioral) and different levels of influence (such as individual and interpersonal). Here, influenced by the "One Health" approach, we propose adding two new levels of influence – interspecies and planetary – to this framework to reflect the interconnected nature of human, animal, and environmental health. Extending the framework in this way will help researchers to create new avenues of inquiry and encourage multidisciplinary collaborations. We then use the One Health approach to discuss how the COVID-19 pandemic has exacerbated health disparities, and show how the expanded framework can be applied to research into health disparities related to antimicrobial resistance and obesity.

**BRITTANY L MORGAN\*, MARIANA C STERN, ELISEO J PÉREZ-STABLE, MONICA WEBB HOOPER AND LAURA FEJERMAN\***

**\*For correspondence:** blmorgan@ucdavis.edu (BLM); lfejerman@ucdavis.edu (LF)

**Competing interest:** The authors declare that no competing interests exist.

## Introduction

Research frameworks are useful because they allow important problems to be tackled from many different directions and perspectives (*Mills et al., 2010*) For the individual researcher, a framework can help with the formulation of research questions and situate their research in a broader context. Research frameworks are also useful to funding agencies: for example, the National Institute on Minority Health and Health Disparities (NIMHD) in the United States employs a research framework to assess the state of current research and to identify gaps and opportunities for funding (*Alvidrez et al., 2019*).

The NIMHD framework has two axes: the vertical axis lists five "domains" that influence minority health and health disparities (biological, behavioral, physical/built environment, sociocultural environment, and healthcare system), while the horizontal axis lists four "levels of influence" (individual, interpersonal, community, and societal). The model builds on a framework for research into health disparities that was developed by the National Institute on Aging, (*Hill et al., 2015*) and also on the socioecological model (*Bronfenbrenner, 1977*; *Kilanowski, 2017*).

The NIMHD encourages using the research framework to examine the interaction of factors from different domains and levels to understand health disparities. Investigators can use the framework while conducting their literature review to map and organize study findings and identify gaps in knowledge. Then, they can formulate research questions and define theories and models evaluating the influence of factors from several levels and domains. For example, the NIMHD has used the framework to evaluate disparities by racial/ethnic category in lung cancer mortality by identifying factors from three domains and levels of influence. Specifically, genetic risk is highlighted as a determinant in the biological domain and at the individual level, while access to quality treatment is an influence in the healthcare system domain and at the community level. Moreover, the framework facilitates considerations of the influence of structural

> ## Box 1. Definitions.
>
> Minority health: Aspects of health and disease among racial and ethnic category minority populations as defined by the US Census.
>
> Health disparity: A preventable difference in health outcomes that adversely affects socially and/or economically disadvantaged populations. These populations include racial and ethnic category minority groups, persons of less privileged socioeconomic status, underserved rural residents, and sexual and gender minorities; all share a social disadvantage in part due to having been subject to discrimination or racism.
>
> One Health: An approach recognizing the health of people is closely connected to the health of animals and our shared environment; the health of one affects the health of all. One Health is a collaborative and multidisciplinary approach to understanding and managing health in a wholistic way that prioritizes ecosystem balance.
>
> Human ecosystem: Combines components of the ecosystem traditionally recognized by ecologists (plants, animals, microbes, physical environmental complex) with the built environment and social characteristics, structures, and interactions of interest.
>
> Research framework: A structure supporting collective scientific endeavours by guiding the development and investigation of a research question and conceptualizing the relationship between relevant factors.

racism on health disparities: differences in state laws, cigarette taxes, social norms, and smoking bans affect racial and ethnic category disparities through the behavioral domain and the societal level (*Alvidrez et al., 2019*). The framework highlights the complex nature of minority health and health disparities, and the need to consider multiple domains and levels of influence when trying to understand these areas and change them for the better (see *Box 1* for definitions of minority health and other terms used in this article).

The importance of merging aspects of environmental health with health equity has long been highlighted by the World Health Organization. In 2008, for example, the Commission on Social Determinants of Health stressed the impact of climate change on the health of individuals and the planet (*Gama and Colombo, 2010*). Increasingly, studies show that biological, cultural and environmental factors – and interactions between these factors – are all relevant to research into health disparities, (*van Daalen et al., 2020*; *Garnier et al., 2020*; *Mueller et al., 2018*) and the COVID-19 pandemic has increased focus on the human-animal-environment interface (*de Garine-Wichatitsky et al., 2020*). However, it is challenging to include the impact of environmental factors and human-animal linkages on health disparities in the NIMHD framework, so we are proposing to expand the framework by adding two new levels of influence – interspecies and planetary – and making use of the "One

Health" approach to thinking about the emergence and prevention of disease. (*CDC, 2020a*) This expansion allows researchers to build on the biological and social determinants of health, and their root causes such as structural racism, already explored in the NIMHD framework. The two new levels of influence explicitly frame questions that link interspecies factors (i.e., the interactions between the human host and their microbiome or animal and human shared infections) and planetary factors (i.e., rising temperatures or global food production practices) with those stemming from biological, social, cultural, and structural and explore their combined impact on health and disease.

## Health disparities and One Health

The One Health approach involves multiple health science professions collaborating to attain optimal health for people, other animals, and the environment (*Schneider et al., 2019*). The One Health approach has primarily been adopted to investigate and prevent the spread of infectious and zoonotic diseases.(*Kelly et al., 2020*; *Schmiege et al., 2020*). For several endemic, zoonotic diseases, One Health approaches targeting animal or environmental reservoirs of infection have proven more equitable than interventions focusing on clinical management of disease, which can be inaccessible for disadvantaged and poor communities. In Latin America, for example, modest investments in mass dog

| The Human Ecosystem* | | | | | | |
|---|---|---|---|---|---|---|
| **Domains of Influence** | **Levels of Influence** | | | | | |
| | **Individual** | **Interpersonal** | **Community** | **Societal** | **Interspecies** | **Planetary** |
| **Biological** | Biological Vulnerability and Mechanisms | Caregiver-Child Interaction Family Microbiome | Community Illness Exposure Herd Immunity | Sanitation Immunization Pathogen Exposure | Shared Pathogenic and Nonpathogenic Microbes | Pathogen Reservoirs Climate Change |
| **Behavioral** | Health Behaviors Coping Strategies | Family Functioning School/Work Functioning | Community Functioning | Policies and Laws | Pet Ownership Dietary Practices | Policies and Laws Global Trade |
| **Physical/Built Environment** | Personal Environment | Household Environment School and Work Environment | Community Environment Community Resources | Societal Structure | Livestock/Wildlife Interactions Land Use/Distribution | Ambient Air Temperature and Pollution Extreme Weather Events Climate Change Effects on Land |
| **Sociocultural Environment** | Sociodemographics Limited English Cultural Identity Response to Discrimination | Social Network Family/Peer Norms Interpersonal Discrimination | Community Norms Local Structural Discrimination | Social Norms Societal Structural Discrimination | Cultural Foods & Practices Food/Meat Production Practices | Migration and Mobility Globalization |
| **Healthcare System** | Insurance Coverage Health Literacy Treatment Preferences | Patient-Clinician Relationship Medical Decision-Making | Availability of Services Safety Net Services | Quality of Care Health Care Policies | Comparative Medicine Holistic Care Animals as Sentinels | Global Health Conservation Medicine Pharmaceutical Waste Medicine Production |
| **Health Outcomes** | **Individual Health** | **Family/Organizational Health** | **Community Health** | **Population Health** | **One Health** | |

**Figure 1.** Proposed expansion of the National Institute on Minority Health and Health Disparities (NIMHD) research framework. The NIMHD research framework includes five domains (rows) and four levels (columns) that influence minority health and health disparities. The proposed expansion of the framework introduces two new levels of influence – the interspecies level and the planetary level (both shaded in grey). The new framework reflects how human health is a product of the human ecosystem, which combines traditionally recognized ecosystem components (plants, animals, microbes, physical environmental complex) with the built environment and social characteristics, structures, and interactions between all these elements. The figure shows examples of some of the factors that are relevant at the intersection between each domain and each level. The origins of the two new levels lie in the "One Health" approach, which recognizes that the health of people is closely connected to the health of animals and our shared environment. The bottom row of the framework demonstrates that health outcomes can also span multiple levels – individual, family and organizational, community, population, and, in the expanded framework, One Health.

vaccinations have effectively prevented rabies-related death in humans and resulted in near elimination of the virus from the community (*Vigilato et al., 2013*). This approach has been more effective, and equitable, than increased expenditures in the administration of post-exposure prophylaxis, which has been emphasized in many Asian countries where there is still high incidence of human rabies cases (*Cleaveland et al., 2017*).

Human health is linked to non-human animal health in many ways: some of these links are direct (e.g., food consumption) and some are indirect (e.g., via the environment), and many of these links are not fully understood (*Davis and Sharp, 2020*; *Wolf, 2015*). Differences in the frequency of human-animal interactions among population groups, (*Rabinowitz and Conti, 2013*) individual and cultural food practices, (*Wolfe et al., 2005*; *Kamau et al., 2021*) livelihood systems,

(*Woldehanna and Zimicki, 2015*) and livestock production practices (*Edwards-Callaway, 2018*; *Ducrot et al., 2008*) could lead to health disparities. These relationships are particularly relevant for populations such as farmworkers, (*Pol et al., 2021*) people experiencing homelessness, (*Hanrahan, 2019*) individuals living in agricultural communities, (*Wing and Wolf, 2000*) and certain racial and ethnic category minorities. For example, individuals with high fish diets have the potential for increased exposure to harmful contaminants influenced by waterway pollutants, marine food webs, and climate change (*Gribble et al., 2016*). The unequal distribution of pollutants is a matter of environmental racism, and this interacts with other social and structural factors disproportionately impacting impoverished communities and communities of color. This may be particularly relevant among Native American

| The Human Ecosystem* | | | | | | |
|---|---|---|---|---|---|---|
| **Domains of Influence** | **Levels of Influence** | | | | | |
| | **Individual** | **Interpersonal** | **Community** | **Societal** | **Interspecies** | **Planetary** |
| **Biological** | Microbiome and Flora Underlying Comorbidities | Family Microbiome | Infectious Disease Prevalence | Wastewater Treatment Safe Water Pathogen Exposure | Shared Pathogenic and Nonpathogenic Microbes | Climate Change Effect on Pathogens Resistance Gene Mobilization Antibiotic Residues |
| **Behavioral** | Hygiene Practices Injection Drug Use | Medicine Sharing Family Hygiene Practices | | Antibiotic Use Laws | Pet Ownership Animal Slaughter Practices Food Preparation | International Trade International Travel Animal Protein Demand |
| **Physical/Built Environment** | Housing Conditions | Household Crowding Occupation/Work Environment | Waste Management | Criminal Justice System and Imprisonment | Livestock Management Manure Management Livestock Employment | Pathogen Reservoirs |
| **Sociocultural Environment** | Sociodemographics Health Literacy and Education | Family/Peer Norms around Antibiotic Use and Hygiene Practices | Social and Cultural Norms around Antibiotic Use and Hygiene Practices | Antibiotic Costs Antibiotic Availability | Culture Foods and Practices Food/Meat Production Practices | Animal Movements Human Migration |
| **Healthcare System** | Antibiotic Use Healthcare Access Insurance Status | Patient-Clinician Relationship Antibiotic Prescribing | Sanitation Infectious Disease Prevention Diagnostic Testing Access | Antibiotic Stewardship Programs Vaccine Availability | Shared Antibiotics in Veterinary and Human Medicine | Antibiotic Drug Pipeline Pharmaceutical Waste Global Antibiotic Resistance Initiatives and Action Plans |
| **Health Outcomes** | **Individual Health** | **Family/Organizational Health** | **Community Health** | **Population Health** | **One Health** | |

**Figure 2.** Expanded framework applied to health disparities research in antimicrobial resistance (AMR). An example of how the expanded framework can be utilized by investigators as they develop their research questions and study designs for research into disparities related to AMR and AMR-related infections. The factors listed under the interspecies and planetary levels of influence are included in a more straightforward and systematic way than they would be in the original NIMHD framework.

and Pacific Islander communities who consume fish at higher rates than other subpopulations (*Washington State Department of Ecology, 2013*). There is limited research in the area, but evidence of elevated blood mercury among these groups (*Hightower et al., 2006*) could lead to health disparities stemming from a complex association between environmental justice, climate change, systemic racism, and this interspecies relationship (*CDC, 2021b*). The NIMHD framework does not easily facilitate consideration of these determinants.

There is a bidirectional relationship between human and environmental health, and the NIMHD framework includes the physical/built environment as a domain of influence. However, the natural environment and climate change can also influence health disparities. For example, extreme weather events or changes to the natural environment can disproportionately impact segments of the population based on geography and access to resources. In the Western part of the United States, the increasing number, size, and intensity of wildfire events

occurring due to extreme heat and drought may disproportionately impact nearby populations through displacement or prolonged exposure to smoke, worsening or creating new populations with health disparities (*US EPA, 2017a*). Further, climate change impacts on health, environmental pollutants, habitat loss, biological diversity, and the distribution of resources, disproportionately impact poor people and populations of color, and can drive or exacerbate health disparities (*van Daalen et al., 2020*; *Huong et al., 2020*; *Hinchliffe et al., 2021*).

As an example of the interconnectedness of plant and human health, consider a mycotoxin called aflatoxin that is produced by common fungi and may play a causative role in hepatocellular carcinoma (*Ramirez et al., 2017*). Hepatocellular carcinoma disproportionately affects Latinos, and aflatoxin commonly affects corn and maize crops, considered a Latin American dietary staple (*Overcash and Reicks, 2021*). Climate change impacts on temperature and drought increase aflatoxin levels in crops and could exacerbate hepatocellular carcinoma disparities among populations

| The Human Ecosystem* | | | | | | |
|---|---|---|---|---|---|---|
| **Domains of Influence** | **Levels of Influence** | | | | | |
| | **Individual** | **Interpersonal** | **Community** | **Societal** | **Interspecies** | **Planetary** |
| **Biological** | Mechanisms of Energy Balance<br>Adipose Tissue Function<br>Microbiome<br>Genetics | Breastfeeding<br>Parent BMI<br>Epigenetics | | Definition of "Obese" State<br>Racism | Exposure to Agrochemicals<br>Adenovirus Exposure<br>Microbiome | CO2 Associated Affect on Crop Nutritional Quality<br>Ambient Temperature and Thermogenesis |
| **Behavioral** | Diet<br>Soda Consumption<br>Physical Activity | Family Income<br>Family and Peer Behavior<br>Social Network | Community Behaviors around Diet and Exercise | Dietary Guidelines<br>Federall Funded Programs<br>Food and Sugar Tax | Dog Ownership<br>Human-Animal Bond<br>Farm Subsidies | Geographic Distribution of Food<br>Extreme Heat Events<br>World Bank Agricultural Investments |
| **Physical/Built Environment** | Personal Food and Physical Activity Environment | Occupational Built Environment<br>Interpersonal Food and Physical Activity Environment | Grocery and Convenience Store Density<br>Fast Food Restaurants<br>Opportunities for Recreation | Structural Inequity<br>Capitalistic Economics | Farm Density<br>Air Quality and Environmental Contamination | Climate Change Impacts on Food Production<br>Urbanization<br>Global Markets |
| **Sociocultural Environment** | Sociodemographics<br>Stress<br>Response to Discrimination | Family/Peer Norms around Diet and Exercise | Ethnic Enclaves<br>Residential Segregation<br>Community Norms around Diet and Exercise | Social Norms and Values around Diet and Physical Activity<br>Discrimination<br>Marketing and Advertisements | Food Production and Farming Practices | Global Demand of Food |
| **Healthcare System** | Insurance Coverage<br>Health Literacy<br>Use of Healthcare Services | Medical Bias and Stigmatization<br>Referral and Services Offered | Preventive Services Offered | Obesity Screening<br>Insurance Programs | Comparative Clinical Research<br>Holistic Medicine | WHO Interventions |
| **Health Outcomes** | **Individual Health** | **Family/Organizational Health** | **Community Health** | **Population Health** | **One Health** | |

**Figure 3.** Expanded framework applied to health disparities research in obesity. An example of how the expanded framework can be utilized by investigators as they develop their research questions and study designs for research into disparities related to obesity. The factors listed under the interspecies and planetary levels of influence are included in a more straightforward and systematic way than they would be in the original NIMHD framework.

relying on the health and safety of those crops (**Kebede et al., 2012**).

The complexity of the linkages humans have with each other, with non-human species, and with the environment should be incorporated with the exploration of social systems and structural racism to understand or model health outcomes and subsequent health disparities (**Craddock and Hinchliffe, 2015**). We believe adding elements of the One Health approach to the NIMHD framework will help expand health disparities research, open new avenues of inquiry for both One Health and health disparities researchers, and promote multidisciplinary collaborations, all of which should lead to broader and more sustainable solutions to health disparities. Using the COVID-19 pandemic as a motivating example, we will use the One Health approach to identify new determinants of health disparities not easily incorporated in the original NIMHD framework. We will then present an expanded research framework and show how this new framework can be used to think about health disparities in the fields of infectious diseases (using antimicrobial resistance as an example) and non-communicable diseases (using obesity as an example).

While the One Health approach and the NIMHD framework are already being used by researchers, the work is conducted in disciplinary silos. One Health researchers may not see how they can help inform and address health disparities or how they can begin incorporating social and structural systems in their work (**Craddock and Hinchliffe, 2015**; **Solis and Nunn, 2021**). Health disparities researchers may not consider the influence of the broader human ecosystem and its interactions with social, cultural, and structural systems. We hope that the expanded research framework will highlight the overlap in their work and stimulate new areas of research and thinking. It is important to note the relationships presented throughout are theoretical and meant to be illustrative of the hypothesis generating potential of the expanded framework, not an assumption of causal relationships or a replacement for exploring the complex social

and structural systems creating health disparities. Further, although the NIMHD research framework focuses on minority health and health disparities in the United States, we will discuss how it might also apply to other countries.

## Health disparities and COVID-19

The disproportionate impact COVID-19 has had on some racial and ethnic category minority populations, and on people with low economic resources, has re-centered conversations concerning health disparities. Those with underlying medical conditions (such as diabetes, heart disease, and chronic lung disease) have also been strongly overrepresented in case severity and fatalities (*Burch and Searcey, 2020*; *Kim et al., 2020*). Marginalized communities already impacted by health disparities are also disproportionately affected by these conditions, increasing the harm caused by COVID-19 (*Kim et al., 2020*; *Lopez et al., 2021*). Several of the factors that contribute to COVID-19 health disparities are accounted for in the NIMHD framework (such as structural racism and its impact on neighborhood and built environment characteristics, healthcare access, occupation and workplace conditions, income, and education *CDC, 2020b*). The One Health approach allows factors not easily accounted for in the NIMHD framework (such as disproportionate access to natural resources like clean air, drinking water, potable water for sanitation and hygiene, and nutritious foods) to be considered (*Garnier et al., 2020*).

The WHO has classified COVID-19 as a zoonotic disease (*WHO, 2020*). While an animal reservoir has not yet been identified, (*Haider et al., 2020*) most zoonotic diseases are thought to enter human populations through a spill-over event originating from human-animal exposure. Unlike some zoonotic diseases, COVID-19 does not require a non-human animal host for pathogen persistence. The disease may have originated in animals and then independently persisted in human populations through respiratory transmission (*Singla et al., 2020*; *Jayaweera et al., 2020*; *Meyerowitz et al., 2021*).

Epidemics of zoonotic origin can be triggered by changes in human and non-human animal reservoirs' interaction dynamics. Environmental, climatic, socio-economic, and habitat or animal abundance changes can modify the probability of human and non-human animal interactions. And as the rate of such interactions increases, so does the rate at which respiratory viruses and other infectious agents evolve and adapt to their new

hosts (*Duffy, 2018*). Understanding the drivers of non-human animal and environmental exposures can build on social and systemic factors by incorporating human-animal linkages and planetary health to deepen our understanding of the COVID-19 pandemic and related health inequities. This improved understanding may strengthen our capacity to prevent and better predict the course of future pandemic threats.

Disease transmission between humans and non-human animals is a primary focus of the One Health approach. Previous coronavirus outbreaks exemplify the ways human interactions with non-human animals can increase viral exposure. Human invasion of the natural environment, contact with livestock, rodents, shrews, or bats, as well as the consumption of rare and wild non-human animals, contributes to infection with viruses we normally would not encounter (*Li et al., 2019*). For COVID-19, exposure to bridge hosts (the species that transmit viruses to humans from their natural reservoirs) may be a predictor of viral exposure early in the outbreak (*Solis and Nunn, 2021*). Hypothesized COVID-19 bridge hosts include animals in animal markets or domesticated animals and livestock (*El Zowalaty and Järhult, 2020*). High rates of exposure to wildlife and livestock is an identified predictor of COVID-19 exposure in China, mainly between poultry and rodents/shrews in living dwellings (*Li et al., 2019*). Viral reservoirs and bridge hosts exposures are associated with less economic resources, inadequate housing and impoverished neighborhoods (i.e., stagnant water, animals residing in dwellings, uncollected trash, and overgrown lots), adding potential mediating factors to determinants of health identified in the NIMHD framework (*Solis and Nunn, 2021*). There are also many aspects of COVID-19 related to minority health and health disparity that we are unable to discuss in detail for reasons of space: these include the impact of the human-animal bond on COVID-19 mental health outcomes, (*Shim, 2020*; *Saltzman et al., 2021*; *Brooks et al., 2018*; *Ratschen et al., 2020*) the influence of environmental factors (notably temperature, humidity, and atmospheric pollution) on morbidity, (*Ratschen et al., 2020*; *Chin et al., 2020*; *Ma et al., 2020*; *Conticini et al., 2020*) and questions related to vaccine equity (*Katz et al., 2021*).

Experts agree that COVID-19 will not be the last pandemic (*Gill, 2020*). Learning from our successes and mistakes during this pandemic will be crucial to prepare appropriately. COVID-19 highlights the importance of considering how

the relationships between humans, non-human animals, and the environment affect minority health and contribute to health disparities. Expanding the NIMHD framework to explicitly include determinants from these domains can help us foresee and better respond to future, global threats by expanding our view of health determinants and drivers of disparities. We believe this is best achieved by incorporating a One Health approach in the NIMHD framework. Recently, a One Health Disparities framework was introduced for zoonotic disease researchers to incorporate the ways the social environment relates to disparities in disease exposure, susceptibility, and expression (*Solis and Nunn, 2021*). Alternatively, we are proposing an expanded NIMHD framework as a way for health disparities researchers to explicitly include determinants stemming from human-animal and human-environmental linkages in their work. The relevance of this expansion will become even more clear heading into the future. As globalization increases, climate change progresses, and population growth continues to strain the relationship between humans, non-human animals, and the environment, thinking about health and health disparities in this holistic way will be essential.

## The expanded framework

Our proposed expansion involves adding two new levels of influence from the One Health approach – the interspecies and planetary levels – to the NIMHD framework (*Figure 1*). The interspecies level covers the interplay, interconnectedness, and interdependencies of humans and non-human species (*Davis and Sharp, 2020*). The definition of interspecies is kept as broad as possible to allow room for new areas of inquiry and flexibility in adapting the framework. The planetary level includes the Earth's natural systems, resources, and biodiversity (*Lerner and Berg, 2017*). Adding these two new levels promotes the evaluation of health disparities mechanisms across disciplines by introducing aspects of the human ecosystem previously missing from the NIMHD framework.

Just as the interpersonal level in the present framework explores human-human interactions, the new interspecies level explores relationships between humans and non-human species across all five domains of influence. Examples of such relationships include microbes shared between humans and non-human animals (i.e., the biological domain), (*Trinh et al., 2018*) pet ownership (behavioral), (*Mueller et al., 2018*) livestock and

wildlife interactions (physical/built environment), (*Hemsworth, 2003*) food production practices (sociocultural environment), (*Edwards-Callaway, 2018*; *Ducrot et al., 2008*) and comparative medicine, a discipline that synergizes health research in human and non-human animal medicine (healthcare system) (*Center for Veterinary Medicine, 2021*).

In the present framework, the societal level of influence includes the presence and actions of governmental and civil society organizations at different levels (such as state, country, or region), (*Alvidrez et al., 2019*) and the new planetary level of influence adds considerations of globalization and impacts of the natural environment. Examples include the effects on minority health and health disparities of climate change (i.e., the biological domain), (*US EPA, 2017b*) global trade (behavioral), (*Friel et al., 2015*) ambient air temperature and pollution (physical/built environment), (*Son et al., 2019*; *Yi et al., 2010*; *Schifano et al., 2013*) migration and mobility (sociocultural environment), (*Castañeda et al., 2015*) global health programs, and the global system for producing medicines and medical devices (healthcare system) (*Newman and Cragg, 2020*). More examples are given in *Figure 1* (but please note that these examples are not meant to be comprehensive or to imply causal inference).

### *Example: Antimicrobial resistance*

We will now show how the expanded framework can be applied to research into health disparities in two areas: antimicrobial resistance (AMR) (*Figure 2*) and obesity. The problem of AMR is primarily driven by antibiotic overuse and misuse in humans, non-human animals, and the environment. Moreover, the interconnected nature of AMR means that it has already been studied by One Health researchers, (*McEwen and Collignon, 2018*; *Robinson et al., 2016*) which makes it a promising candidate for the expanded NIMHD framework. There are documented disparities by racial and ethnic category in antibiotic use and AMR infections in the US (*Olesen and Grad, 2018*; *Hota et al., 2007*; *Iwamoto et al., 2013*). There are also known occupational disparities in AMR pathogen exposure, with those in the agricultural and medical fields being at increased risk (*Fynbo and Jensen, 2018*; *Voss et al., 2005*). Racial/ethnic category minority individuals, such as African American/Black and Hispanic/Latino, are more likely to work in these industries due to a combination of government policies and laws, and the unequal distribution of

income and resources (*Division of Labor Force Statistics, 2020*).

Under the NIMHD framework, the domain/level of influence combinations relevant to AMR disparities could include housing conditions (individual/physical plus built environment), antibiotic sharing among family and peers and access to medications without prescriptions (interpersonal/behavioral), and antibiotic prescribing practices (interpersonal/healthcare system). However, there are other potential sources of health disparities related to AMR that are not accommodated by the NIMHD framework. By encouraging the consideration of interactions between humans and other species through the inclusion of an interspecies level, the expanded framework will enable the identification of some of these factors. One example might be antibiotic sharing between humans and non-human animals (i.e., the healthcare domain). Further, humans and non-human animals share and exchange pathogenic and non-pathogenic bacteria (i.e., the biological domain). A third example is that individuals living in rural and agricultural communities are disproportionately impacted by environmental exposures associated with nearby livestock and manure management practices (i.e., the physical/built environment domain). Additionally, while the evidence on whether livestock raised with antibiotics have higher levels of AMR than those raised without antibiotics is mixed, and we test animal products for antibiotic residues in the US, this is not necessarily true for the rest of the world (*Vikram et al., 2018*; *Van Boeckel et al., 2019*).

Since antibiotic-free meat is often cost-prohibitive for those with low socioeconomic status, consumption of livestock raised with antibiotics in countries with fewer regulations may be important to consider when exploring AMR disparities by income, geography, or race/ethnicity category. The interspecies level of influence explicitly frames human exposure to non-human species in the home, workplace, or environment as relevant determinants of health disparities related to AMR.

Globalization and climate change are also highly relevant when considering AMR, and the planetary level of influence in the expanded framework allows these factors to be considered straightforwardly and clearly. International trade and travel (i.e., the behavioral domain) transfer bacteria and viruses across borders through the import and export of organic materials and human movement. These could be significant determinants to consider as certain immigrant populations in the US frequently travel to their native countries, many of which have a high prevalence of AMR infections (*Nadimpalli et al., 2021*; *Ruppé et al., 2018*). The planetary level is also relevant to the challenge of dealing with pharmaceutical waste. How we manage such waste (i.e., the healthcare system domain), and how such waste is distributed, could be associated with socioeconomic status, putting those of lower status at increased risk for exposure to antibiotic residues (biological domain) and pathogen reservoirs (physical/built environment domain).

The risk factors for AMR are well documented, but our understanding of the combination of factors and interactions that contribute to disparities in AMR is limited (*Holmes et al., 2016*). Including the planetary level of influence in the expanded framework highlights the need for multidisciplinary research that considers the interplay between humans, biodiversity, and the environment (*Venkatasubramanian et al., 2020*). For example, environmental scientists could collaborate on studies evaluating geographic disparities in AMR that explore the possible impact of climate change on the abiotic environment in which people live (i.e., the planetary level/biological domain), and how the effect of this relationship disproportionately impacts segments of the population based on socioeconomic status. While there is no direct evidence of climate change impacts to the abiotic environment leading to disparities in AMR, the expanded framework allows such possibilities to be explored.

### Example: Obesity

Obesity, defined by a body mass index above 30 kg/m$^2$, is a major public health concern due to its contribution to several leading causes of disability and death, including diabetes, heart disease, stroke, osteoarthritis, and cancer (*CDC, 2021a*). Obesity affects people in every country worldwide and the prevalence is increasing, making it classifiable as a pandemic (*Swinburn et al., 2019*). This pandemic results from a number of factors, including urbanization, global trade, and easy access to inexpensive caloric-dense food, (*Swinburn et al., 2019*) and disparities in obesity are evident by geography, gender, age, and socioeconomic status, as well as race and ethnicity categories (*Hill et al., 2014*; *Wang and Beydoun, 2007*). Solutions to the growing obesity pandemic will, therefore, require a multifaceted understanding of all these factors and the interactions between them.

The NIMHD framework facilitated our thinking about long-standing structural racism and how

racial residential segregation, inequitable access to healthy and affordable food, and reduced opportunities for a healthy lifestyle lead to obesity-related health disparities (*Bleich and Ard, 2021*). Behaviors contributing to these disparities may include physical activity and diet (i.e., the individual level/behavioral domain), cheap pricing of low nutritional value foods (the societal level/behavioral domain), targeted marketing or advertising (societal level/sociocultural environment domain) and the distribution of full-service grocery stores, convenience stores, and fast-food restaurants that results from societal level factors such as systemic racism affecting zoning laws and lending practices (community level/physical plus built environment domain) (*Petersen et al., 2019*). We believe the expanded framework (*Figure 3*) can build on these efforts, broaden the scope of health disparities research in obesity, and foster collaboration among researchers from different disciplines.

For example, emerging data shows changes to our microbiome, a non-human species living within us, can affect our health, and obesity (*Feng et al., 2020*). Social, economic, and environmental factors are likely to modify the microbiome over time, resulting in poor health outcomes (*Findley et al., 2016*). Thus, changes to the microbiome may be an important mediator or modifier in studies evaluating the relationship between social factors and obesity among socially disadvantaged communities.

The expanded framework could also guide disparities research in new directions. For example, two zoonotic viruses – avian adenovirus (SMAM-1) and adenovirus 36 (Adv36) – are associated with obesity in both humans and non-human animals (*Ponterio and Gnessi, 2015*). There is some evidence for racial/ethnic category and geographic differences in Adv36 seropositivity, but research in this area (at the intersection between the interspecies level and the biological domain) is limited (*Tosh et al., 2020*; *LaVoy et al., 2021*). Finally, the expanded framework can also facilitate multidisciplinary collaboration. For example, there is a correlation between obesity in dogs and obesity in their owners: (*Bjørnvad et al., 2019*) this suggests that public health professionals could work with veterinarians to disseminate health and educational information regarding pets and owners. This type of approach has the bonus of potentially reaching marginalized populations who may mistrust the medical system, lack health insurance, or participate in alternative medicine practices (and are therefore missed by the traditional routes for disseminating health information). Previous studies have successfully improved access to vital services among some of the highest-risk populations, such as those experiencing homelessness, by leveraging the human-animal bond and providing healthcare services through a One Health model (*Panning et al., 2016*).

The Lancet Commission on Obesity deems climate change effects on health a pandemic, and emphasizes how interconnected it is with the obesity pandemic. Efforts to address disparities in obesity could be strengthened by considering the links between the two pandemics and exploring the influence of planetary factors on disparities. Questions for researchers to address would include: how do increases in atmospheric carbon dioxide affect crop nutritional quality, and how do rising temperatures disproportionately impact the geographic distribution of food crops, thermogenesis, and food insecurity? And what are the impacts of anthropogenic changes to the environment (such as urbanization or increased air pollution) and the increasing global demand for food? With regard to disparities, it will be important to explore how climate change disproportionately impacts individuals of low socioeconomic status through food insecurity, reduced opportunities for physical activity or metabolic processes. For example, rising atmosphere air temperature is associated with less adaptive thermogenesis, the complex metabolic process by which humans burn energy to generate heat. This association is likely to disproportionately harm disadvantaged populations who are more likely to work outdoors and live in hotter urban areas than their more privileged counterparts (*Koch et al., 2021*).

## Conclusions

The NIMHD has developed a flexible and adaptable framework to inform research into minority health and health disparities in the United States. In this article, influenced by the One Health approach, we added two new levels of influence – interspecies and planetary – to the NIHMD framework and demonstrated how the expanded framework could open new avenues of inquiry and encourage multidisciplinary collaborations. We then applied the framework to two examples: AMR and obesity.

While our proposal aims to stimulate novel thinking, we acknowledge the need for more evidence to show that adding a One Health approach can be beneficial to research addressing health disparities. Further, we do not

expect the expanded framework to be relevant and applicable for all fields and topics. Moreover, we accept that many other factors – related to putting collaborations together, obtaining funding for projects, and collecting and analyzing data – must be addressed.

Human health is complex and is influenced by many different factors. By ensuring that our expanded framework includes factors related to interactions between humans and other species, and factors related to globalization and climate change, we believe that it will help ensure that no stone remains unturned in efforts to improve health for all and reduce health disparities.

**Brittany L Morgan** is in the Department of Public Health Sciences and the Center for Animal Disease Modeling and Surveillance (CADMS), Department of Veterinary Medicine, University of California, Davis, United States

blmorgan@ucdavis.edu

http://orcid.org/0000-0002-0280-0904

**Mariana C Stern** is in Departments of Preventive Medicine and Urology, Keck School of Medicine of USC, and the Norris Comprehensive Cancer Center, University of Southern California, Los Angeles, United States

**Eliseo J Pérez-Stable** is in the Office of the Director, National Institute on Minority Health and Health Disparities, National Institutes of Health, Bethesda, United States

http://orcid.org/0000-0002-1577-2738

**Monica Webb Hooper** is in the Office of the Director, National Institute on Minority Health and Health Disparities, National Institutes of Health, Bethesda, United States

**Laura Fejerman** is in the Department of Public Health Sciences and the Comprehensive Cancer Center, University of California, Davis, Davis United States

lfejerman@ucdavis.edu

http://orcid.org/0000-0003-3179-1151

*Author contributions:* Brittany L Morgan, Conceptualization, Visualization, Writing – original draft, Writing – review and editing; Mariana C Stern, Writing – review and editing; Eliseo J Pérez-Stable, Writing – review and editing; Monica Webb Hooper, Writing – review and editing; Laura Fejerman, Conceptualization, Writing – original draft, Writing – review and editing

*Competing interests:* The authors declare that no competing interests exist.

### Funding
No external funding was received for this work.

### Decision letter and Author response
Decision letter https://doi.org/10.7554/eLife.76461.sa1
Author response https://doi.org/10.7554/eLife.76461.sa2

### Data availability
There is no accompanying data for the paper.

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
