## [Decision Letter]

**Decision letter after peer review:**

Thank you for submitting your article "Proposed Incorporation of a One Health Approach in a Framework to Address Minority Health and Health Disparities" to *eLife* for consideration as a Feature Article. Your article has been reviewed by three peer reviewers, and their reports are detailed below. Two of the reviewers have also agreed to be named: Susan Craddock and Alissa Levine.

The reviewers and the *eLife* Features Editor (Peter Rodgers) have discussed the reviews and drafted this decision letter to help you prepare a revised submission.

As you will see from the reports, the reviewers had a number of concerns about how the article was written/structured, as well as some specific concerns about the content of the article. These specific concerns are listed below under the heading Essential Revisions.

The next step will be for a member of the *eLife* Features Team to edit the present version of the article, and then send the edited version to you so that you can check the edits, answer any queries, and address the specific concerns about the content listed below.

Summary:

This manuscript makes a convincing argument for adding two new components (links between human and other life forms, and the global context of health) to the research framework that is used by the National Institute on Minority Health and Health Disparities (NIMHD) to study health disparities. Consideration of these two elements will enable researchers to broaden their perspective and open new channels of scientific inquiry. However, there are a number of points that need to be addressed to make the article suitable for publication.

Essential revisions:

1. The One Health approach needs to be clearly explained near the start of the article.

2. Related to the previous point, please check that you need to use all of the following terms: "One Health concept", "One Health interventions", "One Health approach" and "One Health lens". If any of these terms are equivalent, please use just one of them.

3. In the definitions box, please define healthy disparity without referring to health disparities.

4. Line 207: One of the papers you cite (Solis and Nunn, One health disparities and COVID-19) covers very similar ground to your article. When you cite this article, please say more about its content and how it differs from your article.

5. Lines: 306-307: The way 'racial and ethnic groups' is being used needs some examination. If these individuals are from other countries as the authors suggest, then this can be very different than minority populations born and raised in the US, who typically are poorer than whites, more apt to live in polluted and degraded neighborhoods, and have worse health status. Many who fly back to home countries (South Asian communities are a good example but so too might some going back to Ethiopia or South Africa, etc.) are typically wealthier, better resourced, and have access to quality health care.

6. Performing a study that employs all the elements of the combined framework seems challenging, and possibly unfeasible. Please comment on this. Please also provide examples as to how this combined framework can be used in a single study

7. Given that the One Health and NIMHD frameworks are already being used by researchers, what are the advantages of combining them in a new framework?*Reviewer #1 (Recommendations for the authors):*

This article addresses an important – or potentially important – means of expanding explanations of health inequity. I agree with what the authors are arguing, but they need some pretty significant revisions before this is ready for publication.

In general, authors need to reorganize and tighten this article. The Introduction could flow better, and I don't think the opening definitions of theory versus framework are needed. Opening with some sort of statement on the importance of researching minority health and health equity issues from many perspectives (maybe even invoking covid from the beginning as the latest elucidation of health inequities and their multiple and interacting causes) rather than opening with definitions of framework versus theory would be more compelling.

The current NIMHD framework should first be elaborated, and subsequently authors should point out why it is inadequate to many compromised health situations from covid to AMR to obesity. Then, introduce the OH approach and be clear on what it is. Too often vague terms (resources, environment, food systems) are used rather than actual explanations of what the OH approach is, what it adds, etc. Then introduce all three of your examples. The current organization is confusing and discombobulated.

Finally, the authors' consistent parenthetical inclusion of what category or categories each facet fit into was ultimately distracting. I understand the need to make clear what facets a OH approach brings and in particular which facets are planetary and/or interspecies, but I would ponder dropping the categories – I think if the authors better explain what these facets are in a OH approach, their reader audience can figure out for themselves what is added in terms of the current NIMHD framework.

110-1: Elaborate on what is meant by human health's inextricable links to animal and environmental health. Provide some examples – that way, it makes more sense when you mention evaluating the impact of 'these interactions.' Without elaboration, it is not clear what kind of interactions you are talking about.

115-120: Some of these facets are self-explanatory, but several are not and they need to be explained. Conflicting priorities? Livelihood systems? Interactions in the environment? The examples provided starting at line 122 are good – provide more because this is important to your argument.

Though you subsequently invoke the covid pandemic to illustrate your argument for OH to be added to the current framework, you do need further elaboration on these interactions, providing more examples from how particular facets disproportionately impact marginalized populations. Your wildfire example, for instance, could happen to wealthier populations living in the hills surrounding urban areas in California.

205-207: Explain why exposure to bats, etc. happens more often to marginalized communities.

Is this the only example you provide from covid? Are there not many others you could provide?

209-217: this paragraph seems to be something of a grab-bag of facets, sone of which have more tentative connection to a OH approach – like vaccine equity. Could you take the animal ownership further?

246: This paragraph should go further up when 'interspecies' is first introduced.

In this section on AMR, what about antibiotics use in livestock and the fact that more marginalized communities subsequently consume this meat because livestock raised without antibiotics tends to be more expensive?

306-307: The way 'racial and ethnic groups' is being used needs some examination. If these individuals are from other countries as the authors suggest, then this can be very different than minority populations born and raised in the US, who typically are poorer than whites, more apt to live in polluted and degraded neighborhoods, and have worse health status. Many who fly back to home countries (South Asian communities are a good example but so too might some going back to Ethiopia or South Africa, etc.) are typically wealthier, better resourced, and have access to quality health care.

309-311: This needs further explanation – explain how waste is distributed and that this is one of many examples of environmental racism.*Reviewer #2 (Recommendations for the authors):*

This manuscript makes a convincing argument for incorporating two new components in the framework used to study health disparities: links between human and other life forms, and the global context of health. Consideration of these elements will enable researchers to broaden their perspective and open new channels of scientific inquiry.

1. The authors propose expanding the NIMHD framework to include interspecies and planetary considerations as posited in the One Health approach.

2. The authors draw on contemporary health concerns to convincingly convey the pertinence of incorporating the elements of the One Health approach into research on minority health and health disparities, for both infectious and non-communicable diseases.

3. This theoretically driven manuscript envisions broadening the application of interspecies and planetary frameworks to global contexts; the examples explored (Covid-19, antimicrobial resistance, and obesity) are each of clear global importance. The authors demonstrate how two additional multilayered dimensions enable a critical rethinking of health disparities.

4. The proposed framework will contribute to a more holistic perspective that seeks out interconnectivity among humans, other species, and their environments. This approach proposes a framework -the One Health lens-through which to imagine and engage in research and analysis. The community of researchers who already use the NIMHD framework can add these newly theorized dimensions to the existing framework, thereby broadening the scope of possible questions to be asked.

5. The useful glossary of terms (Definitions) provided at the outset could be expanded to include guidance on the varied terminology used to refer to One Health (One Health Approach, One Health paradigm, in addition to the One Health lens, which presumably provides a research-oriented operationalization). When introducing a novel way of framing research, the accompanying terms should help guide readers and researchers, providing a vocabulary with which to articulate and differentiate its parts.

*Reviewer #3 (Recommendations for the authors):*

This manuscript combines the One Health Model with the framework of the National Institute on Minority Health and Health Disparities (NIMHD) to develop a comprehensive framework viewing health disparities through the lens of socioecological and one health models. The authors rightly emphasize the need to understand the complex interactions of social and environmental factors when evaluating health disparities and the need for multidisciplinary collaborations in these fields. The authors present comprehensive tables in the manuscript indicating how the One Health Model can be combined with the NIMHD model and how this adapted model can be used in health disparities research, with examples from antimicrobial resistance (AMR) and obesity. Yet, modifications are needed to render this manuscript suitable for publication. Below are some of my comments:

1. Some of the published literature advance arguments that are similar to what have been pointed out in this study. For example, an article exists that has combined the One Health Model with the NIMHD constructs; the resulted model is labelled « One Health Disparities ». The authors of this article have explained their One Health Disparities model using the COVID-19 pandemic. (Here is the link to the paper: https://doi.org/10.1093/emph/eoab003). Given that the current manuscript draws on combining the One Health Model and NIMHD in the context of COVID19, the authors need to explain how their paper adds to the literature.

2. While looking at the complex interactions between the elements of the adapted model can definitely be useful, I am concerned about the feasibility of such an effort. I believe one cannot evaluate all these factors and their interactions within one study; only a segment of this fulsome framework can be used in a single study. Given that each of the One Health Model and the NIMHD frameworks are being currently used in separate studies regarding health disparities, AMR and obesity, how can a comprehensive framework with all these factors embedded be helpful?

3. Providing definitions of major concepts included in a research paper is good practice. However, some of the definitions provided are slightly vague. For instance, the authors define health disparities as « a difference in health outcomes adversely affecting populations with health disparities in comparison to a reference group. » This definition is confusing because the authors have used the term « health disparities » in the definition of health disparities. Clear definitions of the concepts included in this manuscript can significantly improve the quality of this work.

4. Some of the interrelationships between the elements of the adapted framework seem to be based on assumptions. While the authors have adequately referenced their statements, it would improve the transparency of this manuscript if the authors provide evidence supporting these assumptions in the text of the manuscript.

---

## [Author Response]

Essential revisions:1. The One Health approach needs to be clearly explained near the start of the article.

The introduction was adjusted to explain the One health approach more thoroughly. Specifically, we introduce One Health at the end of the introduction and discuss the One Health approach, including examples of how it has been leveraged previously, in the first paragraph under Health Disparities and One Health (Line 90).

2. Related to the previous point, please check that you need to use all of the following terms: "One Health concept", "One Health interventions", "One Health approach" and "One Health lens". If any of these terms are equivalent, please use just one of them.

We updated the manuscript with the term One Health approach throughout to reduce confusion.

3. In the definitions box, please define healthy disparity without referring to health disparities.

We have updated the definition of health disparity in the definitions box to,

“A difference in health outcomes adversely affecting socially and/or economically disadvantaged populations that have been exposed to racism or discrimination and have been underserved in health care, in comparison to a reference group.”

4. Line 207: One of the papers you cite (Solis and Nunn, One health disparities and COVID-19) covers very similar ground to your article. When you cite this article, please say more about its content and how it differs from your article.

The Solis and Nunn study use the NIMHD framework to demonstrate that social environments are important to zoonotic disease researchers investigating disparities in rates of transmission, susceptibility, and spillover/spillback in populations. They term these “One Health disparities.” Our paper is expanding the NIMHD framework for researchers outside of One health to begin thinking about the ways human-animal and human-environmental relationships influence health disparities broadly. We addressed the difference between ours and the previous study at line 209-213.

5. Lines: 306-307: The way 'racial and ethnic groups' is being used needs some examination. If these individuals are from other countries as the authors suggest, then this can be very different than minority populations born and raised in the US, who typically are poorer than whites, more apt to live in polluted and degraded neighborhoods, and have worse health status. Many who fly back to home countries (South Asian communities are a good example but so too might some going back to Ethiopia or South Africa, etc.) are typically wealthier, better resourced, and have access to quality health care.

We thank the reviewer for pointing out the nuances of this argument and allowing us to clarify our intent. The idea behind this determinant is that individuals traveling more frequently to foreign countries are at increased risk of contact with and contracting AMR bacteria endemic in those areas. They also transfer and carry those pathogens back to the US and their communities. To address and clarify this, we changed ‘racial and ethnic groups’ to immigrant populations at line 286. Further, we added why frequent travel to home countries may present increased risk at line 287.

In addition, many working class immigrants do travel to their home countries especially within the Americas. The proximity of Mexico, the seasonal nature of employment in some cases, and the continued strong bonds with family and place drive this need to return. Immigrants are frequently earning incomes that are several times higher than their families in the home country. Finally, the self-identified construct of race and ethnicity does have global application even if the categories will vary according to region.

6. Performing a study that employs all the elements of the combined framework seems challenging, and possibly unfeasible. Please comment on this. Please also provide examples as to how this combined framework can be used in a single study

We agree performing a study that employs all the elements of the combined framework would likely be impossible. The intent behind the framework is to help guide researchers in developing multilevel and multidimensional studies, understand what the domain and level of influence of their study is at, and to consider what factors may be influencing their findings (even if they can’t measure or quantify it). It also helps understand the broader landscape in which a study is situated. We clarified this intent in the introduction of the paper at line 51 and an example of how the framework has been used follows at line 55.

7. Given that the One Health and NIMHD frameworks are already being used by researchers, what are the advantages of combining them in a new framework?

This is an excellent question. Briefly, we discuss in the paper the benefits of incorporating the One Health approach to the NIMHD framework (addressing health disparities holistically, opening new avenues of inquiry, and promote multidisciplinary collaborations – line 140). We also added a sentence at line 150 to 153 to highlight how the work by One Health researchers and health disparities researchers is conducted in disciplinary silos and how the expanded framework may highlight the overlap in their work and stimulate new ideas.